# Genomic-Thermodynamic Phase Synchronization: Maxwell’s Demon-like Regulation of Cell Fate Transition

**DOI:** 10.3390/ijms26104911

**Published:** 2025-05-20

**Authors:** Masa Tsuchiya, Kenichi Yoshikawa, Alessandro Giuliani

**Affiliations:** 1SEIKO Life Science Laboratory, SEIKO Research Institute for Education, Osaka 540-6591, Japan; 2Faculty of Life and Medical Sciences, Doshisha University, Kyotanabe 610-0394, Japan; keyoshik@mail.doshisha.ac.jp; 3Environment and Health Department, Istituto Superiore di Sanitá (Italian NIH), 00161 Rome, Italy; alessandro.giuliani@iss.it

**Keywords:** cancer cell fate decision, chromatin remodeling, critical point, genome intelligence, genomic-thermodynamic phase synchronization, higher-order mutual information, Maxwell’s demon activation, non-equilibrium information thermodynamics, self-organization, time-series whole-expression data

## Abstract

Dynamic criticality—the balance between order and chaos—is fundamental to genome regulation and cellular transitions. In this study, we investigate the distinct behaviors of gene expression dynamics in MCF-7 breast cancer cells under two stimuli: heregulin (HRG), which promotes cell fate transitions, and epidermal growth factor (EGF), which binds to the same receptor but fails to induce cell-fate changes. We model the system as an open, nonequilibrium thermodynamic system and introduce a convergence-based approach for the robust estimation of information-thermodynamic metrics. Our analysis reveals that the Shannon entropy of the critical point (CP) dynamically synchronizes with the entropy of the rest of the whole expression system (WES), reflecting coordinated transitions between ordered and disordered phases. This phase synchronization is driven by net mutual information scaling with CP entropy dynamics, demonstrating how the CP governs genome-wide coherence. Furthermore, higher-order mutual information emerges as a defining feature of the nonlinear gene expression network, capturing collective effects beyond simple pairwise interactions. By achieving thermodynamic phase synchronization, the CP orchestrates the entire expression system. Under HRG stimulation, the CP becomes active, functioning as a Maxwell’s demon with dynamic, rewritable chromatin memory to guide a critical transition in cell fate. In contrast, under EGF stimulation, the CP remains inactive in this strategic role, passively facilitating a non-critical transition. These findings establish a biophysical framework for cell fate determination, paving the way for innovative approaches in cancer research and stem cell therapy.

## 1. Introduction

Dynamic criticality in gene expression, characterized by shifts in global expression profiles initiated by changes in specific gene subsets, plays a pivotal role in regulating genome dynamics and driving critical transitions [1]. The delicate balance between order and chaos enables genomic regulation that is both flexible and stable [2,3].

Our previous studies demonstrated that this balance underlies both precise cellular responses to external signals and fundamental processes such as cell differentiation and embryonic development. These dynamics are central to understanding cell fate changes. Our key findings are summarized below:**Emergent Self-Organization within a Genome Attractor:** Self-organized critical (SOC) control of genome expression is quantified using the normalized root mean square fluctuation (*nrmsf*) of gene expression, which delineates distinct response domains (critical states). Large gene groups exhibit coherent stochastic behavior (CSB), converging around centers of mass (CM); in our approach, these CMs are treated as unit masses, that act as attractors [4,5,6]. The whole genome CM is the main genome attractor (GA), while local critical states serve as subcritical or supercritical attractors. Cyclic fluxes among these attractors create an open thermodynamic “genome engine” that regulates genome expression [7,8].**CP as a Central Organizing Hub**: A specific set of genes, termed the critical point (CP), exhibits bimodal singular behavior based on the *nrmsf* metric. The CP functions as a central hub, spreading expression variability changes across the entire genome and driving critical transitions. When the system deviates from homeostasis, a state change at the CP propagates perturbations throughout the genome, demonstrating SOC in genomic regulation [9].**Modeling the Genome Engine as a Dynamical System**: The genome engine can be modeled as a one-dimensional dynamical system [10]. The CM’s expression level at time ε_t_ represents “position”, its first difference (ε_t+1_ − ε_t_) represents “momentum”, and its second difference (ε_t+1_ − 2ε_t_ + ε_t−1_) acts as an “effective force” influencing energy changes. This framework, grounded in stochastic thermodynamics [11,12,13,14,15], illustrates how the genome maintains a near balance of influx and outflux while dynamically interacting with its environment.**Transition-Driven Switching of Cyclic Flows**: Critical transitions in the genome engine reverse cyclic expression fluxes [7,8]. These transitions are linked to structural changes, such as the bursting of peri-centromeric domains (PADs) in MCF-7 cancer cells, which affect chromatin folding and enable dynamic genomic regulation [3,16,17].**OMG–CP–GA Network Synchronizing CP and GA for Cell-Fate Change:** The OMG–CP–GA network governs cell-fate transitions by coordinating interactions among Oscillating-Mode Genes (OMGs), the CP, and the GA. OMGs, identified by high scores on the second principal component (PC2, with PC1 representing the equilibrium gene expression profile) [3], modulate synchronization between the CP and GA. This synchronization maintains genome-wide balance and triggers critical transitions, leading to coordinated shifts in genome expression and chromatin remodeling [9].**Advancements over Classical Self-Organized Criticality Models**: Unlike classical SOC (cSOC) models, which involve state transitions from subcritical to supercritical toward a critical attractor [18,19,20,21], our SOC model features a dynamic CP that actively induces state changes to guide cell-fate transitions. This mechanism enables the genome to adaptively regulate itself in response to stimuli [6,7,8].**Universality Across Biological Systems:** The SOC control of genome expression is demonstrated in distinct biological systems, including cell differentiation and embryonic development: HRG- and EGF-stimulated MCF-7 cells [22], atRA- and DMSO-stimulated HL-60 cells [23], Th17 cell differentiation [24], and early embryonic development in mice [25] and humans [26]. These findings highlight the robustness and universality of the dynamic criticality model [6,7,8,27].

Building on these findings and utilizing our dynamic approach, this study aims to:Elucidate how the non-equilibrium thermodynamics of open systems governs genome-wide expression;Distinguish between effective and non-effective dynamics in cell-fate transitions;Propose a unified framework that integrates the genome engine mechanism (as a dynamical system) with genomic thermodynamics.

We analyze temporal gene expression data from MCF-7 cancer cell lines stimulated with epidermal growth factor (EGF), which promotes proliferation without altering cell fate, and heregulin (HRG), which commits cells after a relatively rapid decay of an inital gene expression perturbation to differentiate [3,17,22]. By modeling these two processes as open stochastic thermodynamic systems, characterized by exchanges of heat and matter (e.g., gases, nutrients, and wastes) with their surroundings and also by inherent randomness, we capture the dynamic and non-deterministic nature of genome regulation influenced by both external and internal factors.

To assess differences in genome expression dynamics, we use Shannon entropy and mutual information as measures of disorder and predictability. Shannon entropy, rooted in information theory, quantifies unpredictability in biological processes and relates to entropy production in nonequilibrium systems, linking informational disorder to energy dissipation. This framework explains how biological systems maintain order and perform work under stochastic, far-from-equilibrium conditions.

In contrast, mutual information is essential for understanding feedback, control mechanisms, thermodynamic efficiency, and information flow [28,29,30,31,32]. A key historical perspective on the interplay between information and thermodynamics comes from Maxwell’s demon, a thought experiment introduced by physicist James Clerk Maxwell [33].

Maxwell’s demon sorts molecules to reduce entropy in a closed system, seemingly violating the second law of thermodynamics. However, extensive studies [30,31,34,35,36,37,38,39,40,41,42] have demonstrated that Maxwell’s demon is not merely a paradox but a key concept for understanding the relationship between information and thermodynamics. These studies show that the demon’s ability to measure, store, and process information incurs a thermodynamic cost, thereby preserving the second law.

Toyabe et al. (2010) [38] first experimentally demonstrated the conversion of information into usable work via real-time feedback control, effectively realizing a modern Maxwell’s demon with a colloidal particle system. Building on this foundation, Sagawa and Ueda (2012) [30,31] and Parrondo et al. (2015) [11] formalized a general thermodynamic framework in which the demon operates as a memory through three sequential phases: initialization (or preparation), measurement, and feedback, each carrying a well-defined energetic cost. Subsequent non-quantum experiments have continued to validate and extend this paradigm [43,44,45]. Far from being merely a metaphor, Maxwell’s demon embodies a functional principle, underscoring the profound connection between information processing and thermodynamic behavior in nonequilibrium open systems, including within biological phenomena.

### The Overview and Objectives of Our Study Are as Follows

Understanding complex gene expression networks requires accurate estimation of information-theoretic metrics such as Shannon entropy and mutual information, which quantify disorder, predictability, and interdependencies. These metrics are challenging to estimate due to the stochastic nature of gene expression and the complexity of underlying networks.

To address this challenge, we exploit coherent stochastic behavior (CSB) through a convergence-based approach [4,5,6]. This enhances the estimation of the CP (critical point) location along the axis of gene expression variability (Section 2.1) and ensures the robust identification of CP genes by grouping them based on their expression variability (Section 2.2). Moreover, this approach improves the robustness of information-thermodynamic metrics, including entropy and mutual information, which quantify the dynamics of genomic regulation (Section 2.3). This methodology allows us to characterize the genomic mechanisms distinguishing stimuli that do not promote cell-fate change (EGF stimulation) from those that drive cell-fate transitions (HRG stimulation).

Our study focuses on the following objectives:**Open Thermodynamic Modeling and Phase Synchronization** (Section 2.4): Develop a data-driven approach to model an open stochastic thermodynamic system of MCF-7 cell lines and investigate how order–disorder phase synchronization between the CP and the whole expression system (WES) regulates genome expression under EGF and HRG stimulation.**Higher-Order Mutual Information** (Section 2.4): Investigate complex nonlinear interdependencies in networks within the WES by identifying higher-order mutual information.**CP as Maxwell’s Demon-like Rewritable Chromatin Memory** (Section 2.5 and Section 2.6): Examine the CP’s role as a Maxwell’s demon-like rewritable chromatin memory, clarifying its function in entropy regulation and information flow. This deepens our understanding of genomic thermodynamics and is further supported by the Appendix A.**Gene Expression Variability as a Proxy for Chromatin Remodeling Dynamics** (Section 2.7): Investigate whether gene expression variability serves as a proxy for underlying chromatin remodeling dynamics by analyzing experimental evidence.

In Section 3, we expand on our findings and explore their potential implications for controlling the dynamics of cancer cell fate through three key perspectives: (1) the order parameter of phase synchronization as a bridge between dynamical systems and non-equilibrium open thermodynamics; (2) autonomous genome computing as a conceptual framework; and (3) future perspective toward genome intelligence (GI). Finally, the main insights are summarized in Section 4.

## 2. Results

### 2.1. Normalized Temporal Variability of Gene Expression as a Metric of Self-Organization and a Proxy for Chromatin Flexibility

To identify the critical point (CP) genes, which exhibit critical behavior, we introduce a metric parameter that quantifies the self-organization of time-series whole-genome expression data obtained from both microarray and RNA-Seq datasets (see methodological details in [10]). This metric parameter is defined by the root mean square fluctuation (*rmsf_i_*), representing the standard deviation of a gene’s expression levels over time, calculated as follows:(1)$${rmsf}_{i}=\sqrt{\frac{1}{T}\sum_{j=1}^{T}{\left(\varepsilon_{i}\left(t_{j}\right)-\langle \varepsilon_{i}\rangle \right)}^{2}}$$
where *ε_i_*(*t_j_*) is the expression level of the *i*th gene at a specific cell state or experimental time point *t_j_*, $\langle \varepsilon_{i}\rangle$ is the average expression level of the *i*th gene over all time points, and *T* is the total number of cell states or experimental time points.

To compare gene expression variability across different biological systems [7,8], we normalize the *rmsf_i_* value by the maximum *rmsf* observed in the dataset, resulting in the normalized *rmsf* (*nrmsf_i_*). For each gene expression value measured across experimental time points, we first compute *nrmsf_i_* and then take its natural logarithm; in our study, “expression” specifically refers to this logarithm, *ln*(*nrmsf_i_*). This transformation captures scaling response behaviors in SOC control [5,6,10,27] and noise reduction.

For microarray data, which rarely contain zeros, this approach works effectively. In contrast, RNA-Seq data, often characterized by numerous zero values, require adding a small random noise to zeros, performing ensemble averaging, and then applying the logarithmic transformation to preserve coherent stochastic behavior (CSB) [10].

The use of *ln(nrmsf)* as a ranking parameter is biologically justified because gene expression variability, as quantified by *ln(nrmsf)*, reflects chromatin flexibility. Greater variability indicates that chromatin is more accessible and dynamically regulated. In HRG-stimulated MCF-7 cancer cells undergoing a cell fate transition, principal component analysis (PCA) reveals that principal component scores accurately predict each gene’s *ln(nrmsf)*. In particular, the second and third principal components (PC2 and PC3) capture chromatin structural flexibility associated with chromatin remodeling during the critical transition that guides cell fate [3]. Additional experimental evidence presented in Section 2.7 further supports the use of *ln(nrmsf)* as a proxy for chromatin flexibility [16,17].

The CP genes, identified by a critical range of *ln(nrmsf)* values, serve as regulatory hubs that orchestrate genome-wide transitions [7,8,9]. In this study, we demonstrate that the activation of CP genes at a critical transition drives coherent chromatin remodeling and explore how this process parallels Maxwell’s demon–like behavior (see Section 2.5 and Section 2.6).

### 2.2. Development of Information-Thermodynamic Analysis via Identification of CP Genes Using a Metric Parameter

The whole-genome mRNA expression dataset is analyzed using *ln(nrmsf)* as the metric parameter. Genes are first sorted in descending order of *ln(nrmsf)* and then grouped into clusters large enough to capture genome-wide averaging behavior of collective expression dynamics. Coherent stochastic behavior (CSB) in gene expression [4,5,6] emerges when the sample size (*n*) exceeds 50 [46]. As the sample size increases, the group’s center of mass (CM) with unit mass converges to a constant value, driven by the law of large numbers. This convergence reflects a nearly balanced inflow and outflow of expression flux within the whole expression system (WES) [10] (see more in Section 2.4). To grasp such types of thermodynamic behavior, each group needs to contain more than 200 genes.

Figure 1 illustrates the data-processing workflow for time-series whole gene expression, which underpins the information-thermodynamic analysis (ITA) and computation of the associated information-thermodynamic metrics. This framework establishes a data-driven approach for modeling the gene expression dynamics of MCF-7 cells as an open stochastic thermodynamic system (see Section 2.3). The objective is to investigate how phase synchronization between the critical point (CP) and the whole expression system (WES) regulates genome-wide expression in response to EGF and HRG stimulation. Particular focus is on identifying distinct information-thermodynamic differences between conditions that induce cell-fate change (HRG) and those that do not (EGF), despite both ligands signaling through the same receptor.

**Figure 1 ijms-26-04911-f001:**
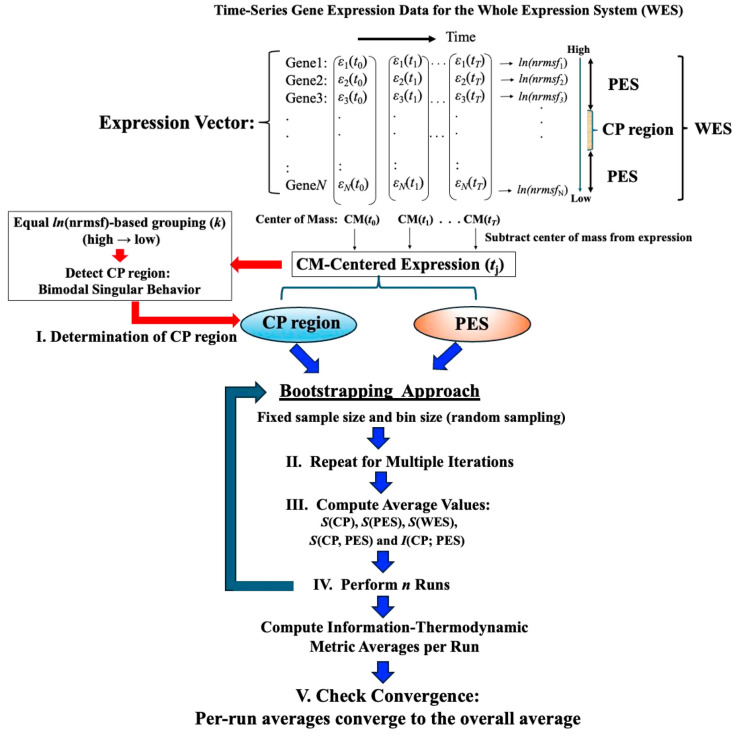
**Workflow for computing information-thermodynamics metrics from time-series gene expression data**. This figure illustrates a five-step workflow for assessing information-thermodynamics metrics from the time-series gene expression data of MCF-7 cells. The process begins by constructing an *N*-dimensional expression vector (*N* = 22,277, corresponding to the number of genes on the array) for the whole expression system (WES) at each time point, using normalized raw expression data to perform background adjustment and reduce false positives (see Section 4). For each gene, the *nrmsf* value is then computed and transformed using the natural logarithm, together with the natural logarithm of the gene expression values (see Section 2.1 for details). The log-transformed gene expression values, after subtraction of the time-dependent center of mass (CM), are then sorted in descending order based on their *ln(nrmsf)* values. From this sorted data: (**I**) the critical point (CP) region is determined by identifying a specific set of gene expressions that exhibit bimodal singular behavior, using equal-sized *ln(nrmsf)*-based grouping with *k* = 40 groups (see Figure 2). To estimate robust CP entropy, the CP region, defined as the range of *ln(nrmsf)* values exhibiting bimodal singular behavior, is standardized by adjusting its original interval by 0.2 units (see Figure 2). This adjusted range includes both CP genes and adjacent edge genes influenced by CP dynamics, thereby providing a sufficient gene set. (**II**) The peripheral expression system (PES) is defined as the remaining portion of the WES that interacts with the CP region to establish an open stochastic thermodynamic model of MCF-7 cell lines (Figure 3). (**III**) Based on the CP and PES expression vectors, bootstrapping is performed using fixed parameters (random sample size = 1000; bin size = 30), while varying the number of bootstrap iterations (see Figure 4), and the average values of the resulting metrics are computed. (**IV**) The entire bootstrapping process (run) is repeated *n* times (*n* = 10 in this study), with averaged metric values computed for each individual run. (**V**) Convergence is confirmed when the average absolute difference between successive bootstrap iterations (500 and 1000 iterations for mutual information; 200 and 500 iterations for entropy) across all time points falls below 10^−3^. Definitions of the computed metrics (e.g., *S*(CP), *I*(CP; PES)) are provided in the main text.

As illustrated in Figure 2, the critical point (CP) region is identified through temporal changes in overall expression along the *ln(nrmsf)* metric; genes within this region exhibit distinct bimodal singular temporal behavior. This characteristic behavior serves as a basis for distinguishing the peripheral expression system (PES) from the whole expression system (WES). In HRG-stimulated MCF-7 cells, the CP region and PES comprise 3846 and 18,431 genes, respectively, while in EGF-stimulated MCF-7 cells, they comprise 4033 and 18,244 genes, respectively. Hereafter, the CP region is referred to simply as the CP genes, unless a distinction between the region and the specific gene set is necessary.

**Figure 2 ijms-26-04911-f002:**
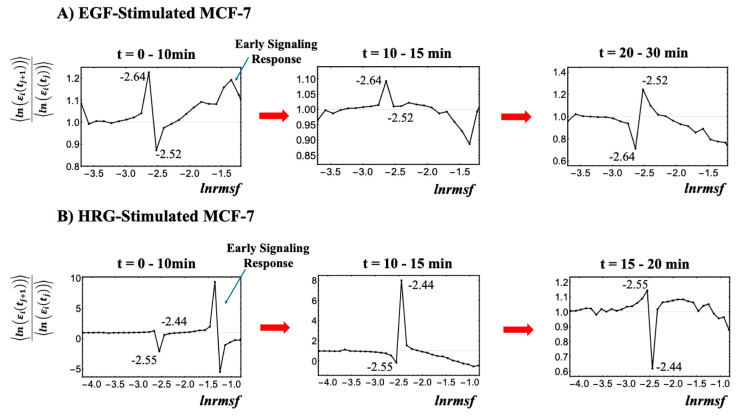
**Identification of the critical point (CP) region with distinct response domains**: (**A**) EGF-stimulated MCF-7 cells and (**B**) HRG-stimulated MCF-7 cells. This figure elaborates on step (I) from the workflow presented in Figure 1. The entire gene expression dataset (*N* = 22,277 genes) is sorted by *ln(nrmsf)* and divided into 40 equal-sized groups, excluding those with fewer than 200 genes due to insufficient convergence at the high and low extremes of the distribution. Since *ln(nrmsf)* values are time-independent, they are plotted on the *x*-axis. The *y*-axis represents the ratio of group ensemble averages across different time points, <*ln*(ε_i_(t_j+1_))>/<*ln*(ε_i_(t_j_))>, in which the CP region distinctly emerges with bimodal singular behavior. A black solid dot represents the natural logarithm of the ensemble average of the *nrmsf* value, <*nrmsf*_i_> for the *i*^th^ group on the *x*-axis, while the *y*-axis shows the ratio of the group’s average expression between consecutive time points, <*ln*(ε_i_(t_j+1_))>/<*ln*(ε_i_(t_j_))>. Under EGF stimulation (**A**), CP genes exhibit bimodal critical behavior with peaks ranging from −2.64 to −2.52 *ln(nrmsf)*, defining the CP region of −2.7 < *ln(nrmsf)* < −2.5 (4033 genes). Under HRG stimulation (**B**), the peaks range from −2.55 to −2.44 *ln(nrmsf)*, defining the CP region of −2.6 < *ln(nrmsf)* < −2.4 (3846 genes). The CP region effectively separates high-variance from low-variance responses, thereby highlighting distinct regulatory domains within the genome. For details regarding the early signaling response, refer to Section 2.4.

### 2.3. Open Stochastic Thermodynamic Model of MCF-7 Cell Lines

This section presents a stochastic thermodynamic model of MCF-7 breast cancer cell lines, focusing on the interaction between the critical point (CP) gene region and the rest of the whole gene expression system (WES). The remaining genes in WES are collectively referred to as the peripheral expression system (PES).

We adopt an open thermodynamic framework in which cells continuously exchange heat and substances (gases, nutrients, and waste products) with their environment. This exchange influences gene expression dynamics. By analyzing temporal entropy variations within the system, we gain insights into how entropy constrains the organization and stability of genomic regulation. This approach highlights open thermodynamic principles that govern these genomic-level interactions.

As illustrated in Figure 3, both the CP genes and the PES exchange energy with the external environment (OUT). We analyze an ensemble of CP genes and the PES within the cancer cell population, examining their entropy and interactions to uncover the open thermodynamic properties governing the temporal evolution of the system.

**Figure 3 ijms-26-04911-f003:**
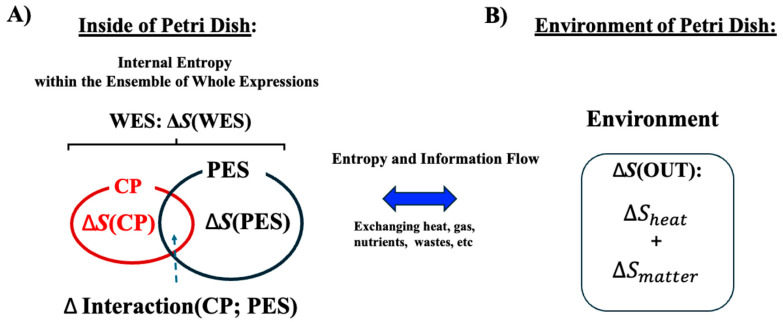
**Open stochastic thermodynamic model setup for MCF-7 cell lines**. This schematic depicts an open thermodynamic system for MCF-7 cell lines, where heat, gases, nutrients, and wastes are continuously exchanged with the environment. The system includes the whole expression system (WES), which comprises the critical point (CP) genes and the remaining genes, collectively referred to as the peripheral expression system (PES). Each component has its own entropy: *S*(WES), *S*(CP), and *S*(PES). (**A**) Within the system, the total entropy change of WES (Δ*S*(WES)) is decomposed into changes in the CP (Δ*S*(CP)), the PES (Δ*S*(PES)), and their interaction term (ΔInteraction(CP; PES)). This decomposition enables investigation of how these interaction terms relate to the mutual information Δ*I*(CP; PES). (**B**) Entropy exchange with the environment (Δ*S*(OUT)) consists of contributions from heat (Δ*S*(heat)) and matter exchange (Δ*S*(matter)). We apply the coherent stochastic behavior (CSB) method to estimate entropy and mutual information, demonstrating consistent convergence behavior despite the inherent randomness of individual genes (see Figure 4).

Figure 4 illustrates that, with a fixed sample size (30), increasing the number of sampling repetitions via bootstrapping enables robust estimation of information-thermodynamic metrics. This approach leverages CSB to achieve convergence of average values across samplings, thereby stabilizing the estimates and revealing the thermodynamic principles that govern the genome’s dynamic regulatory mechanisms. Additionally, the second thermodynamic condition is considered, accounting for entropy flow between the cell line and its external environment (beyond the confinement of the Petri dish), as described below.

(1): **Random Sampling**: We repeatedly perform bootstrap random sampling with replacement on gene sets from the CP, PES, and WES datasets.

(2): **Calculating Entropy**: For each sample, we calculate the Shannon entropy using the following equation:(2)$$S\left(X\left(t_{j}\right)\right)=-\sum_{i}P\left(x_{i}\left(t_{j}\right)\right)ln\left(P\left(x_{i}\left(t_{j}\right)\right)\right)$$
where *S*(*X*(*t_j_*)) is dimensionless; *X*(*t_j_*) represents CP(*t_j_*), PES(*t_j_*), or WES(*t_j_*); and *P*(*x_i_*(*t_j_*)) is the probability of gene *x_i_* being expressed at *t_j_*.

Note that regarding Figure 4, we observed that varying the bin sizes used to calculate the probability distributions leads to consistent offsets in the resulting information-thermodynamic metrics (e.g., entropy), with larger bin sizes producing higher values. This behavior is exactly as expected for entropy as a proper state function. We chose a bin size of 30 for the probability distributions, following the square root rule (bin size ~ $\sqrt{n}$, where *n* is the sample size), as described in [47].

(3): **Relating Entropies and Higher-Order Mutual Information**: we compute the joint entropy *S*(CP, PES) at *t_j_* as follows:(3)$$S\left(CP\left(t_{j}\right),PES\left(t_{j}\right)\right)=-\sum_{x_{i\in CP,y_{j}\;\in PES}}P\left(x_{i}\left(t_{j}\right),y_{j}\left(t_{j}\right)\right)ln\left(P\left(x_{i}\left(t_{j}\right),y_{j}\left(t_{j}\right)\right)\right).$$
where *P*(*x_i_*(*t_j_*), *y_j_*(*t_j_*)) is the joint probability of the CP gene *x_i_* and PES gene *y_j_* being simultaneously expressed at *t_j_*. To estimate the joint probability distribution between CP and PES genes, we construct a two-dimensional joint frequency table at each time point using a bootstrapping approach and a fixed bin size of 30 (Figure 1).

The mutual information (MI) between CP and PES, *I*(CP: PES), is given as follows:(4)$$I\left(CP\left(t_{j}\right);PES\left(t_{j}\right)\right)=S\left(CP\left(t_{j}\right)\right)+S\left(PES\left(t_{j}\right)\right)-S\left(CP\left(t_{j}\right),PES\left(t_{j}\right)\right)$$

The joint entropy *S*(CP, PES) can also be expressed as internal entropy as follows:(5)$$S_{internal}=S\left(CP\left(t_{j}\right),PES\left(t_{j}\right)\right)=S\left(CP\left(t_{j}\right)\right)+S\left(PES\left(t_{j}\right)\right)-I\left(CP\left(t_{j}\right);PES\left(t_{j}\right)\right)$$
where the mutual information is subtracted to remove shared information, leaving only the unique and combined uncertainties of the two components. From this point onward, the time *t_j_* will be omitted for simplicity, except when explicitly necessary.

**Figure 4 ijms-26-04911-f004:**
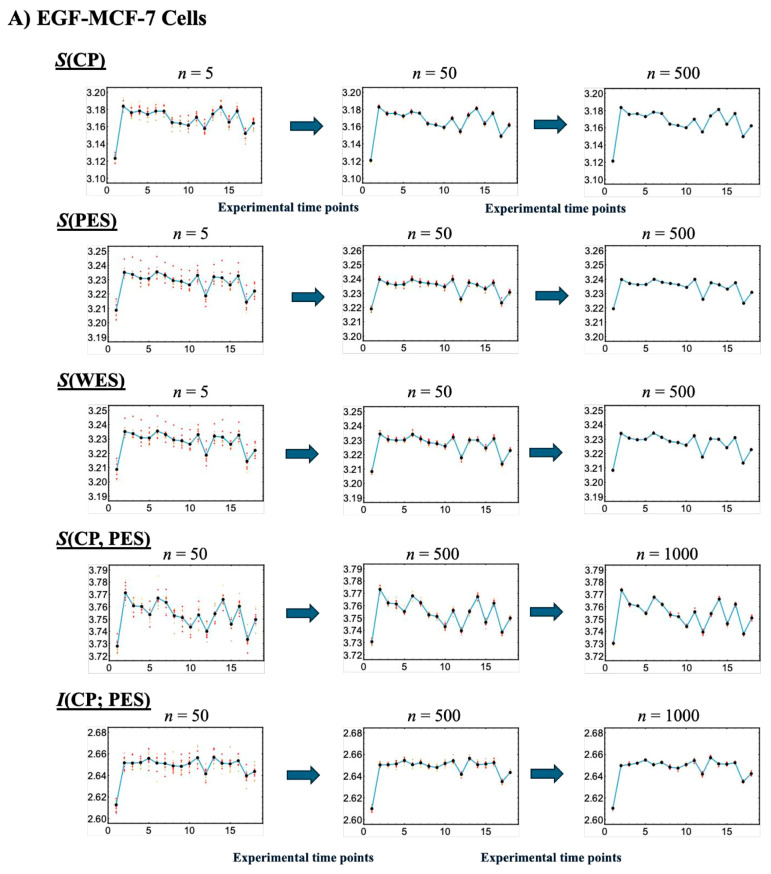
**Convergence-based approach for robust estimation of information-thermodynamic metrics.** Panels (**A**) EGF-stimulated and (**B**) HRG-stimulated MCF-7 cells. This figure illustrates the details of step V in the workflow outlined in Figure 1, focusing on the convergence assessment. This approach enables robust estimation of key information-thermodynamic metrics: entropy of the CP (*S*(CP)), WES (*S*(WES)), and PES (*S*(PES)); as well as joint entropy (*S*(CP, PES)) and mutual information (*I*(CP; PES)). Each row shows the progression of convergence for a specific metric, with estimates computed via bootstrapping (sample size = 1000; bin size = 30), following the square root rule for the sample size as described in [47]. The number of iterations increases across columns to illustrate convergence dynamics: entropy metrics are computed for 5, 50, and 200 iterations, while joint entropy and mutual information for 50, 500, and 1000 iterations, due to their higher dimensionality. To assess convergence, the entire bootstrapping run is repeated 10 times under each condition, and the resulting metric values at each time point are plotted. Convergence is visually evaluated by the reduction in scatter among the 10 averaged runs, as highlighted by red solid circles approaching the overall average (black solid circles). Joint entropy and mutual information typically exhibit slower convergence because they require estimating two-dimensional joint frequency distributions, unlike the one-dimensional distributions used for entropy. The *x*-axes represent experimental time points, while the *y*-axes denote the corresponding metric values.

Note that the WES consists of both CP genes and the remaining WES genes (PES). The dimensionless entropies *S*(WES), *S*(CP), and *S*(PES) are calculated from their respective gene sets using the same bootstrapping procedure (Figure 1). As shown in Figure 5, *S*(WES) differs from *S_i_*_nternal_ (Equation (5)) at all time points, indicating that the standard mutual information formula (Equation (4)) does not fully capture all dependencies in MCF-7 cancer cells:(6)$${S\left(WES\right)\;\neq S}_{internal}=S\left(CP,PES\right)$$

By replacing *S*(CP, PES) in Equation (4) with *S*(WES), the net MI including higher-order terms, *I*_net_(CP; PES), is defined as follows:(7)$$I_{net}\left(CP;PES\right)=\left\{S\left(CP\right)+S\left(PES\right)\right\}-S\left(WES\right)$$

Therefore, from Equations (4) and (7), the higher-order MI, *I*_*high*_(CP; PES) can be defined and observed as positive for the entire experimental time:(8)$$\begin{array}{cc} I_{high}\left(CP;PES\right) & =I_{net}\left(CP;PES\right)-I\left(CP;PES\right)\\ & =\left\{S\left(CP\right)+S\left(PES\right)\right\}-S\left(WES\right)-\left\{S\left(CP\right)+S\left(PES\right)-S\left(CP;PES\right)\right\}\\ & =S\left(CP;PES\right)-S\left(WES\right)>0 \end{array}$$

Higher-order mutual information (MI) [48,49,50,51] extends beyond standard MI by capturing collective and network-level effects, including overlapping gene functions, feedback loops, and nonlinear interdependencies that simple pairwise correlations cannot fully explain. In Figure 5, a positive higher-order MI (*I*_*high*_ > 0) under both EGF and HRG stimulation indicates that the total system entropy *S*(WES) is less than the combined CP and PES entropies. This suggests the presence of redundancy (overlapping information) and synergy (collective effects exceeding pairwise contributions) in the MCF-7 genomic network; see partial information decomposition [52]. These findings support the presence of backup mechanisms, overlapping functions (redundancy), and emergent collective regulation (synergy) in a typical biological genomic network.

In Figure 6, under both EGF and HRG stimulations, the net MI, *I*_net_(CP; PES)) are nearly equivalent to the CP entropy S(CP):(9)$$I_{net}\left(CP;PES\right)\;\sim \;S\left(CP\right)+0.007\;\mathrm{for EGF stimulation}$$
and(10)$$I_{net}\left(CP;PES\right)\;\sim \;S\left(CP\right)+0.010\;\mathrm{for HRG stimulation}$$

Since the net MI exceeds *S*(CP) slightly, it reflects the positive higher-order MI contribution. This result clearly demonstrates that CP genes drive coordinated transitions between ordered and disordered phases between the CP and PES. Moreover, it provides a thermodynamic framework to explain how state changes in the CP can drive large-scale ‘genome avalanches,’ as demonstrated through expression flux analysis [9].

(4): **Second Law of Thermodynamics:** In our open system, the whole expression system comprising CP genes and PES, the entropy production (*σ*) quantifies the irreversible processes occurring within the system. Compliance with the second law of thermodynamics mandates that entropy production [53] must be non-negative (*σ* ≥ 0), representing the irreversibility inherent in processes like metabolic reactions, molecular interactions, and active transport mechanisms.

Dimensionless entropy production (*σ*) is mathematically defined as the difference between the total change in the WES entropy (ΔS(WES)) and the entropy flow exchanged with the external environment of the culture medium (ΔS(OUT):(11)$$\sigma =\Delta S\left(WES\right)-\Delta S\left(OUT\right)\geq 0$$

The change in entropy of the outside environment, Δ*S*(OUT) is given as follows:(12)$$\Delta S\left(OUT\right)=\Delta S_{heat}+\Delta S_{matter}$$
where(13)$$∆S_{heat}=\frac{Q}{T}$$
represents the entropy change due to heat (*Q*) exchange at a temperature (*T* = 37 °C), and Δ*S*_matter_ is the net entropy exchange associated with the inflow and outflow of substances:(14)$$\Delta S_{matter}=\sum_{out}n_{out}\cdot s_{out}-\sum_{in}n_{in}{\cdot s}_{in}$$
where *n_in_* and *s_in_* represent the molar quantity and molar entropy of inflowing substances, and *n_out_* and *s_out_* for outflows.

Cells generate heat through metabolic activities (e.g., ATP synthesis, substrate oxidation), exchange nutrients (e.g., glucose, amino acids), and waste products (e.g., CO_2_, lactate) with the environment.

Therefore, we obtain the second law condition for the system:(15)$$\sigma =\Delta S\left(WES\right)-\left(\frac{Q}{T}+\Delta S_{matter}\right)\;\geq 0$$

Internal entropy production *σ* must be non-negative (*σ* ≥ 0) to satisfy the second law of thermodynamics. This ensures that all irreversible processes within the cell culture contribute to an overall increase in entropy.

### 2.4. Genomic-Thermodynamic Phase Synchronization Between CP Genes and the Whole Expression System

The genomic-thermodynamic phase synchronization dynamics for EGF-stimulated and HRG-stimulated MCF-7 cells are summarized below:

**EGF-stimulated MCF-7 cells (non-cell fate change):** EGFR activation drives entropy changes in CP and PES genes within the first 10 min. Thereafter, information-thermodynamic metrics fluctuate around their temporal mean value through synchronized order–disorder phases, guided by net mutual information dynamics that scale with CP entropy. The WES maintains a dynamic balance of entropy and information flux.

**HRG-stimulated MCF-7 cells (cell fate change):** The PES exhibits stronger phase synchronization with the CP than under EGF stimulation, as revealed by net mutual information dynamics. A critical transition at 10–30 min activates the CP as a Maxwell demon, driving feedback to the PES and triggering a genomic avalanche. This produces a pulse-like perturbation in all information-thermodynamic metrics.

A key difference between EGF and HRG stimulation is the emergence of this critical transition via activation of a Maxwell’s demon-like mechanism (Section 2.5 and Section 2.6; see also the conclusive remarks in Section 2.7).

Further details are provided below:
(1)In EGF-stimulated MCF-7 cells:
**(A)** **Dissipative Thermodynamic Synchronization of Order–Disorder Phases**In EGF-stimulated MCF-7 cells, EGFR activation triggers the MAPK/ERK pathway for gene regulation and the PI3K/AKT pathway for survival, driving proliferation within the first 10 min [22]. This activation induces marked changes in the entropy of the CP region genes and PES, as evidenced by the rise in net mutual information, *I*_net_(CP; PES)), during the initial 0–10 min (Figure 5A).The increase in *I*_net_(CP; PES) reflects nonlinear interactions between the CP and PES that expand both of their accessible state microspaces, such as different gene expression patterns, protein states, or regulatory interactions, which indicates active exchanges of heat and matter with the environment. As the CP’s complexity grows, its nonlinear influence on the PES strengthens, promoting greater integration between these subsystems. Furthermore, the phase synchronization between the CP and PES, characterized by coordinated shifts between ordered and disordered phases (Figure 6A), yields a high Pearson correlation (*r* = 0.90). This synchronization is fundamentally driven by *I*_net_(CP; PES), which combines standard mutual information with positive higher-order MI (Section 2.3) and scales with *S*(CP).**(B)** **Dynamic Equilibrium Underpinning Coherent Stochastic Behavior (CSB)**Figure 5A shows that *S*(WES) remains nearly constant, fluctuating only between 3.208 and 3.234 around its mean of 3.227—a total range of 0.026 units. *This balance of entropy and information flux preserves overall stability and supports coherent stochastic behavior (CSB).* This observation aligns with genome engine dynamical systems [7,9], suggesting a biophysical basis for the law of large numbers, where collective system behavior remains stable despite the inherent stochasticity of individual molecular interactions.(2)In HRG-stimulated MCF-7 cells:**Activation of the CP as a Maxwell Demon**In contrast to the large initial disorder changes induced by EGF in the first 0–10 min, HRG stimulation triggers pulse-like order–disorder transitions across all information-thermodynamic metrics. These transitions are driven by rising net MI, *I*_net_(CP; PES), with both *S*(CP) and *S*(PES) increasing during the 10–20 min period (Figure 5B). This suggests more active exchanges of heat and matter (entropy exchange and information flow) with the environment under HRG, thereby strengthening phase synchronization. The pulse-like critical transition aligns with findings from PCA analysis [3] and expression flux analysis [7,8]. It involves the CP, acting as a Maxwell demon (see detailed analysis in later sections), inducing feedback to the PES and orchestrating a genomic avalanche that drives cell-fate change.From a biological standpoint, within 5–10 min window, HRG stimulation, distinct from EGF, activates the ERK/Akt pathway, leading to a ligand-specific, biphasic induction of AP-1 complex components (e.g., c-FOS, c-JUN, FRA-1) and the transcription factor c-MYC [22,54,55].By 15–20 min, the above-sketched biochemical cascade produces a pulse-like, genome-wide perturbation marked by the activation of oscillating-mode genes (OMGs), as shown by expression flux analysis [9]. These genes emit large expression fluxes to both the CP and the genome attractor (GA), facilitating their synchronization within the OMG–CP–GA network.At 60 min, this critical transition is followed by a peak in c-MYC expression [16,17], resulting in extensive gene amplification. Beyond this point, entropy changes exhibit steady, damped fluctuations, as observed also in both PCA and expression flux analyses. This indicates balanced large-scale energy flows that sustain cellular homeostasis.

These findings on EGF- and HRG-stimulations demonstrate that changes in entropy and net mutual information effectively serve as indicators of fundamental cellular processes, including the activation of signaling pathways, the occurrence of critical transitions, and the synchronization of genome-wide expression. Adopting this thermodynamic perspective enriches our understanding of how different stimuli orchestrate complex regulatory networks within cells.

**Figure 5 ijms-26-04911-f005:**
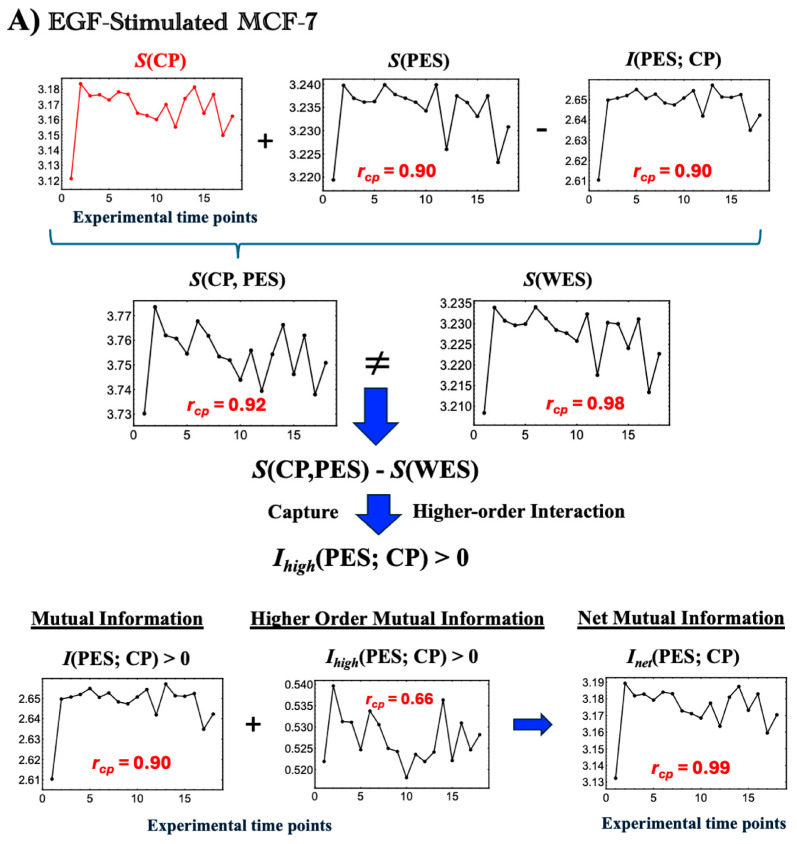
**Thermodynamic phase synchronization and higher-order mutual information in MCF-7 cells.** (**A**) EGF stimulation; (**B**) HRG stimulation. Using a bootstrapping approach (500 iterations for entropy and 1000 iterations for mutual and joint entropy, as described in Figure 4), we calculated the entropies *S*(WES), *S*(CP), and *S*(PES). These figures highlight the following three key points. (1) According to the standard mutual information framework, the relation *S*(WES) = *S*(CP) + *S*(PES) − *I*(CP; PES), where *S*(WES) = *S*(CP, PES) is expected to hold. However, the observed non-zero difference between *S*(WES) and the joint entropy *S*(CP, PES) indicates additional positive higher-order mutual information, *I_high_*(CP; PES) > 0. This implies that the net mutual information *I*_net_(CP; PES) comprises both standard and higher-order components (refer to Section 2.3). (2) The high temporal Pearson correlation (***r***_cp_) observed between CP entropy and the net mutual information *I*_net_(CP; PES), as well as *S*(PES) and *I*(CP; PES), demonstrates CP-PES phase synchronization. This synchronization is more pronounced under HRG stimulation than under EGF stimulation, as indicated by higher temporal correlation with *S*(CP) (see more details in Figure 6 and related biological regulations in Section 2.4). (3) In HRG stimulation, a critical transition occurs within the 10–15–20 min range during the CP-PES phase synchronization, associated with chromatin remodeling and the activation of a Maxwell demon-like mechanism (see Section 2.5, Section 2.6 and Section 2.7). The *x*-axis indicates time points (0, 10, 15, 20, 30, 45, 60, 90, 120, 180, 240, 360, 480, 720, 1440, 2160, 2880, and 4320 min), and the *y*-axis represents the measured values (entropy and mutual information in dimensionless units). See the main text for further details.

**Figure 6 ijms-26-04911-f006:**
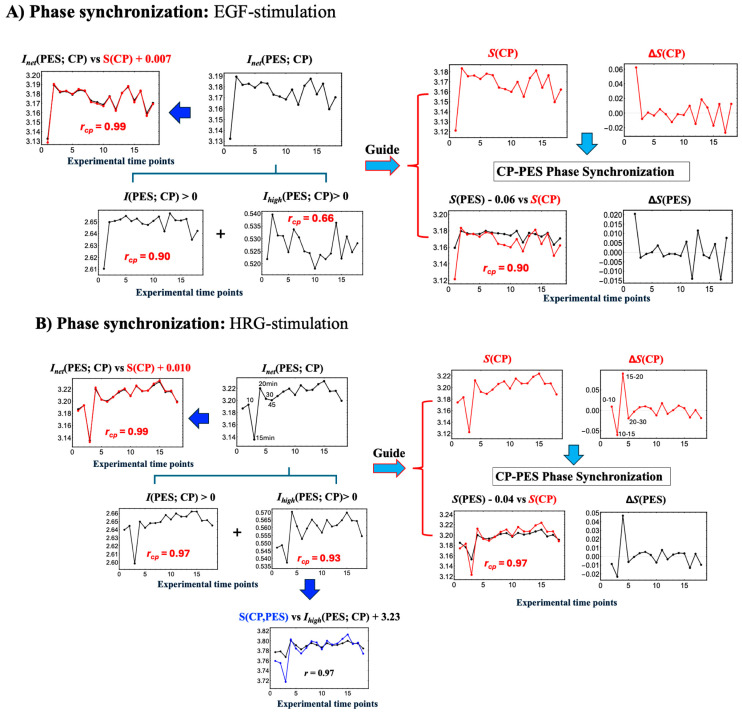
**Net MI as proxy for thermodynamic CP–PES phase synchronization**. (**A**) EGF stimulation; (**B**) HRG stimulation. These figures show that thermodynamic CP-PES phase-synchronization is driven by the net mutual information, *I*_net_(CP; PES) = *I*(CP; PES) + *I*_*high*_(CP; PES), where *I*_*high*_ captures nonlinear contributions from higher-order interactions. In both conditions, *I*_net_ closely follows CP entropy *S*(CP) with a Pearson correlation of *r*_cp_ = 0.99; under EGF stimulation, *I*_net_ ≈ *S*(CP) + 0.007, and under HRG stimulation, *I*_net_ ≈ *S*(CP) + 0.010 (upper left panels; black line: *I*_net_; red line: *S*(CP)). For HRG stimulation, *I*_*high*_(CP; PES), scales with the joint entropy as *S*(CP, PES) ≈ *I*_*high*_(CP; PES) + 3.23 (***r***_cp_ = 0.97), indicating a substantial nonlinear contribution to CP–PES coupling. A pulse-like increase in *I*_net_ at 10–30 min signals chromatin remodeling consistent with a Maxwell’s demon-like rewritable chromatin memory, as detailed in Section 2.5, Section 2.6 and Section 2.7.

### 2.5. CP Acts as Maxwell’s Demon in Early HRG Signaling Response

Unlike EGF, which induces early disorder (0–10 min), HRG triggers a distinct critical transition, synchronizing ordered and disordered phases across all information-thermodynamic metrics from 10 to 20 min. As shown in Figure 5B and Figure 6B, the CP acts as a Maxwell demon [11], regulating information flow and maintaining adaptable chromatin states (see Section 2.6 for details).

The CP, functioning as a Maxwell’s demon, orchestrates gene expression and cell fate through the following three distinct phases:**Phase 1: Preparation (10–15 min)**: The CP synchronizes with the PES to minimize the entropies of both CP and PES genes, establishing their initial states. Biologically, this synchronization sustains the upregulation of AP-1 complex genes and coordinates the activation of their downstream targets, ensuring the proper assembly and functionality of AP-1 components (see Section 2.4).**Phase 2: Measurement (15–20 min)**: Synchronization between the CP and PES initiates the measurement process, leading to the largest increases across all information-thermodynamic metrics, including CP and PES entropy, as well as standard, higher-order, and net MI.**Phase 3: Feedback and Critical Transition (20–30 min)**: Utilizing the net MI obtained in Phase 2, the CP reorganizes and provides feedback to the PES, triggering a critical transition around 20 min in the WES [3,7,8]. This feedback-driven reorganization reduces system entropy, driving the system into a new ordered state.

Further details are provided below:**Phase 2: Measurement Phase (15–20 min)**

**1.** 
**Largest Increases in CP Entropy, PES Entropy, and Net MI**


As shown by the 15- and 20-min data points in Figure 5B (or Figure 6B), the CP genes acquire the most information about the PES during the measurement phase. Processing and storing this information raises the CP entropy, owing to the associated thermodynamic costs. This phase exhibits the largest increases in both standard and higher-order MI, which drive net MI upward from its 15-min minimum. Notably, the surge in higher-order MI underscores strong nonlinear interdependencies between the CP and PES.

Figure 6B shows that higher-order MI is strongly correlated with the joint entropy *S*(CP, PES) (*r* = 0.97). As total uncertainty, *S*(CP, PES), increases, more states become accessible, expanding the system’s global search space and amplifying structured interactions between the CP and PES via the exchange of information and entropy fluxes with the environment (see note below). The positive higher-order MI reflects both synergy (emergent collective information) and redundancy (overlapping information) (see Section 2.3), enabling the CP to encode information about PES despite rising global uncertainty. This open, non-equilibrium thermodynamic phase synchronization, led by the CP genes, is a hallmark of complex systems, where localized organization persists within high-entropy environments, supporting robust information processing.

Note: In the scenario described by Parrondo et al. (2015) [11], the entropy of the measured system (here, the PES) decreases as it loses uncertainty to the measuring agent (the CP), thereby increasing its internal order. However, in our system, the PES is part of an open environment that continuously exchanges energy. During phase synchronization, rather than experiencing a net entropy reduction from information loss, the PES absorbs energy—whether as heat, free energy, or chemical substrates—from its environment. This absorbed energy offsets the entropy decrease that would typically result from the measurement, leading instead to an overall rise in *S*(PES). Consequently, this energy influx sustains phase synchronization between the CP and PES, thereby supporting the feedback mechanism essential for Maxwell’s demon function.

The synchronized metrics in Phase 2 lay the foundation for the feedback process in Phase 3, driving the system’s critical transition to a new cell state.


**Phase 3: Feedback and critical transition**


**1.** 
**Initiation of Critical Transition (20–30 min):**


During this interval, the CP’s decreasing entropy induces a corresponding drop in PES entropy through phase synchronization, facilitated by entropy exchange and information flow with the environment. Through feedback, the CP utilizes net mutual information to drive a critical transition, revealing the thermodynamic costs of its activity. Concurrently, declines in both standard MI and higher-order MI reduce net MI, suggesting that the CP leverages information from Phase 2 to guide the PES through the critical transition. Notably, the timing of this CP-driven feedback at 20 min aligns with a genome-wide transition, as revealed by expression flux analyses of the genome engine mechanism [9] and principal component analysis (PCA) [3].

**2.** 
**Stabilization into New States After 30 min:**


After 30 min, fluctuations in *S*(CP) and *S*(PES) occur, and net mutual information suggests that both CP and PES are settling into stable, reorganized states. Notably, between 30 and 45 min, the bimodal pattern of the CP dissipates, and by 60–90 min, a new CP emerges, indicating the establishment of a new state (see Section 2.6).

### 2.6. Maxwell’s Demon Functioning as Rewritable Chromatin Memory

J. Krigerts et al. (2021) [16] recently investigated experimentally critical transitions in chromatin dynamics during early differentiation in HRG-stimulated MCF-7 breast cancer cells, focusing on pericentromere-associated domains (PADs). They identified two key critical transitions:**First Critical Transition (15–20 min)**: Following HRG treatment, PADs undergo a “burst”, dispersing from their clustered state near the nucleolus. This dispersal coincides with the activation of early response genes, such as c-fos, fosL1, and c-myc, which initiate the differentiation process. During this phase, repressive chromatin structures unravel, and active chromatin regions become more accessible, marking a significant step in genome reorganization.**Second Critical Transition (Around 60 min):** The second transition involves further chromatin alterations, including increased transcription of long non-coding RNAs (lncRNAs) from PADs. This supports additional chromatin unfolding and genome rewiring, which are essential for establishing and maintaining differentiation, ultimately determining a stable and functional cell fate.

In this section, we explore the information-thermodynamic mechanisms underlying these two critical transitions in chromatin dynamics. To achieve this, we analyze the metric parameter *ln(nrmsf)* (Equation (1)), which indicates the self-organization of the whole expression system (WES) into distinct response domains (critical states) separated by a critical point (CP) (see Figure 2; see [10]). This metric quantifies the temporal variability of gene expression as a time-independent measure and serves as a proxy for chromatin remodeling dynamics.

Zimatore et al. (2021) [3] showed that *ln(nrmsf)* effectively captures time-based variability in gene expression and that principal component (PC) scores from temporal whole-expression data accurately predict each gene’s *ln(nrmsf)* (see Section 2.1). Notably, during the critical transition at 10–30 min, a one-order-of-magnitude increase in the variance explained by the second principal component (PC2), which represents the primary direction of displacement from the genome attractor (GA), coincides with peri-centromeric body splitting and chromatin unfolding, thereby exposing new genomic regions to polymerases [16] (see Section 2.7). Because PC2 dynamics mirror *ln(nrmsf)* values and Shannon entropy is computed directly from the *ln(nrmsf)* distribution, *ln(nrmsf)* serves as a quantitative proxy for entropy changes, thereby mechanistically linking chromatin remodeling to thermodynamic shifts.

As illustrated in Figure 7, raw gene-expression data are binned into *m* discrete *ln(nrmsf)* intervals. Each interval defines a logical state—a coarse chromatin conformation. The distribution of expression values inside an interval defines the associated physical states, recording gene-level fluctuations within that chromatin setting. This hierarchy matches Maroney’s (2009) [56] extension of Landauer’s principle: the entropy of logical states quantifies large-scale chromatin reorganization, whereas the residual entropy of physical states captures gene-specific regulatory fluctuations within each domain.

Robust entropy estimates are obtained with the convergence protocol detailed in Figure 1 and Figure 4. For each time point *t*_j_, the per-gene entropy contribution −*p*(*x*_i_(*t*_j_))*ln*(*p*(*x*_i_(*t*_j_)) is iteratively assigned to its corresponding *ln(nrmsf)* bin, and bootstrapping is repeated until the convergence criterion is satisfied. This iterative procedure incorporates every expression value at the logical level and produces numerically stable, reproducible entropy estimates.

By integrating logical and physical components, our ITA links abrupt structural transitions, such as the sharp increase in variance captured by PC2 and the switching of the genome engine (Figure 8 in [10]), to their effects on gene-expression landscapes. The analysis therefore clarifies how chromatin dynamics modulate transcriptional states and ultimately steer cell-fate decisions.

Note that the workflow in Figure 7 analyzes chromatin dynamics using a fixed number of chromatin states (*ln(nrmsf)* bins). The two-dimensional approach in Appendix B extends this framework by simultaneously incorporating logical and physical states, enabling an integrated assessment of both large-scale and local entropy contributions.

**Figure 7 ijms-26-04911-f007:**
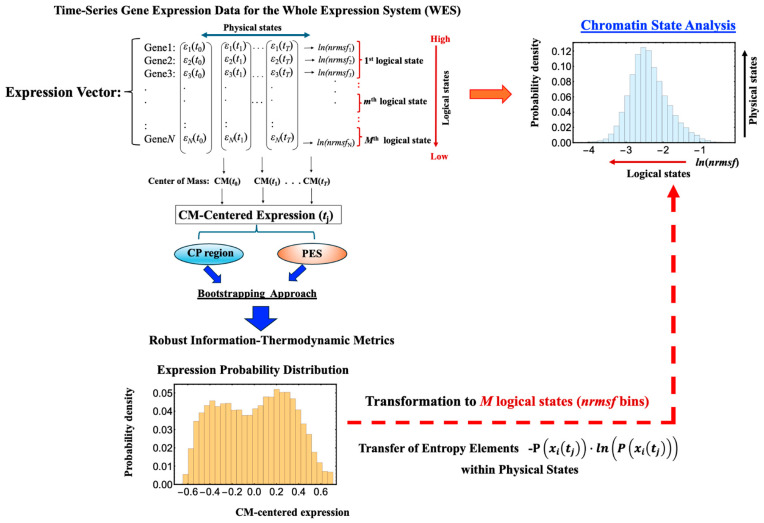
**Information-theoretic workflow for converting gene expression to *ln(nrmsf)* probability distributions**. This schematic outlines the procedure for transforming gene expression probability distributions into *ln(nrmsf)* probability distributions. In the *ln(nrmsf)* probability distribution, the *x*-axis represents logical states, defined as *ln(nrmsf)* bins (fixed) that quantify large-scale transitions in chromatin states (see Section 2.6 and Section 2.7), while the *y*-axis captures physical states, reflecting the fine-scale variability of gene expression within each chromatin domain. The workflow begins with temporal gene expression data from the WES, which encompasses both the CP region and the PES. This approach ensures the robustness of our information-thermodynamic metrics (see Figure 1). Per-gene entropy components, given by −*p*(*x_i_*(*t_j_*))*ln*(*p*(*x_i_*(*t_j_*))), are computed at each time point *t_j_* and then mapped onto the *ln(nrmsf)* bins, thereby converting detailed gene expression profiles into logical states that highlight major chromatin state transitions. This transformation allows us to capture dynamic variability at both the logical (large-scale) and physical (local) levels. For additional details on this process, refer to Figure 8.

Figure 8 depicts the temporal variations in Shannon entropy linked to chromatin dynamics across the WES for EGF- and HRG-stimulated MCF-7 cells:**EGF-stimulated MCF-7 cells** (Figure 8A): This panel shows the temporal changes in Shannon entropy linked to chromatin dynamics in EGF-stimulated MCF-7 cells. Within the specified range (−2.64 < *ln(nrmsf)* < −2.52; Figure 1), a CP-like (critical point-like) bimodal pattern in entropy changes occasionally appears, suggesting transient critical behaviors in chromatin organization. During these intervals, positive entropy changes (ΔS > 0) are associated with chromatin unfolding, while negative entropy changes (ΔS < 0) are associated with chromatin folding.Although these CP-like bimodal patterns emerge intermittently, such as during the 0–10 and 20–30 min windows, they dissolve and reappear without inducing a sustained critical transition or altering cell fate. This implies that the conditions needed to activate a Maxwell’s demon-like mechanism are not fully met. Notably, these chromatin fluctuations are driven by CP–PES phase synchronization, underscoring the role of thermodynamic interactions in maintaining dynamic chromatin states despite the absence of a full critical transition.**HRG-stimulated MCF-7 cells** (Figure 8B): The scenario of Maxwell’s demon-like chromatin memory progresses through distinct phases. During the initial 0–10 min, a CP-like bimodal pattern (−2.55 < *ln(nrmsf)* < −2.44; Figure 1) emerges. At 10–30 min, a global fold–unfold coherent transition occurs around *ln(nrmsf)* = −3.11 to −3.04, involving Maxwell’s demon-like behavior (see Section 2.5). This phase acts as a rewritable chromatin memory, reflecting changes in double-well potential profiles (Figure 9). At 30−45 min, the CP pattern dissolves, likely preparing the system for a new stable genomic state of the WES. At 60–90 min, CP formation reappears, suggesting a cell-fate change consistent with the second transition described by Krigerts et al. (2021) [16]. Beyond 90 min, CP-like bimodal patterns continue to form and dissolve intermittently, indicating sustained chromatin adaptation under HRG stimulation.

Using the *ln(nrmsf)* metric, Shannon entropy reveals chromatin folding and unfolding dynamics. EGF stimulation repeatedly dissolves and reinitiates CP bimodal pattern formations without inducing cell-fate changes, whereas HRG stimulation facilitates stable CP formation and cell fate transitions driven by Maxwell’s demon-like chromatin memory at 10–30 min. As shown in Figure 9, by measuring genomic states, orchestrating entropic and informational dynamics, and reorganizing chromatin, the CP actively encodes (stores), decodes (reads), and re-encodes (rewrites) chromatin configurations. This process aligns with the concept of memory storage and rewriting within a dynamically regulated, thermodynamically controlled genomic landscape.

**Figure 8 ijms-26-04911-f008:**
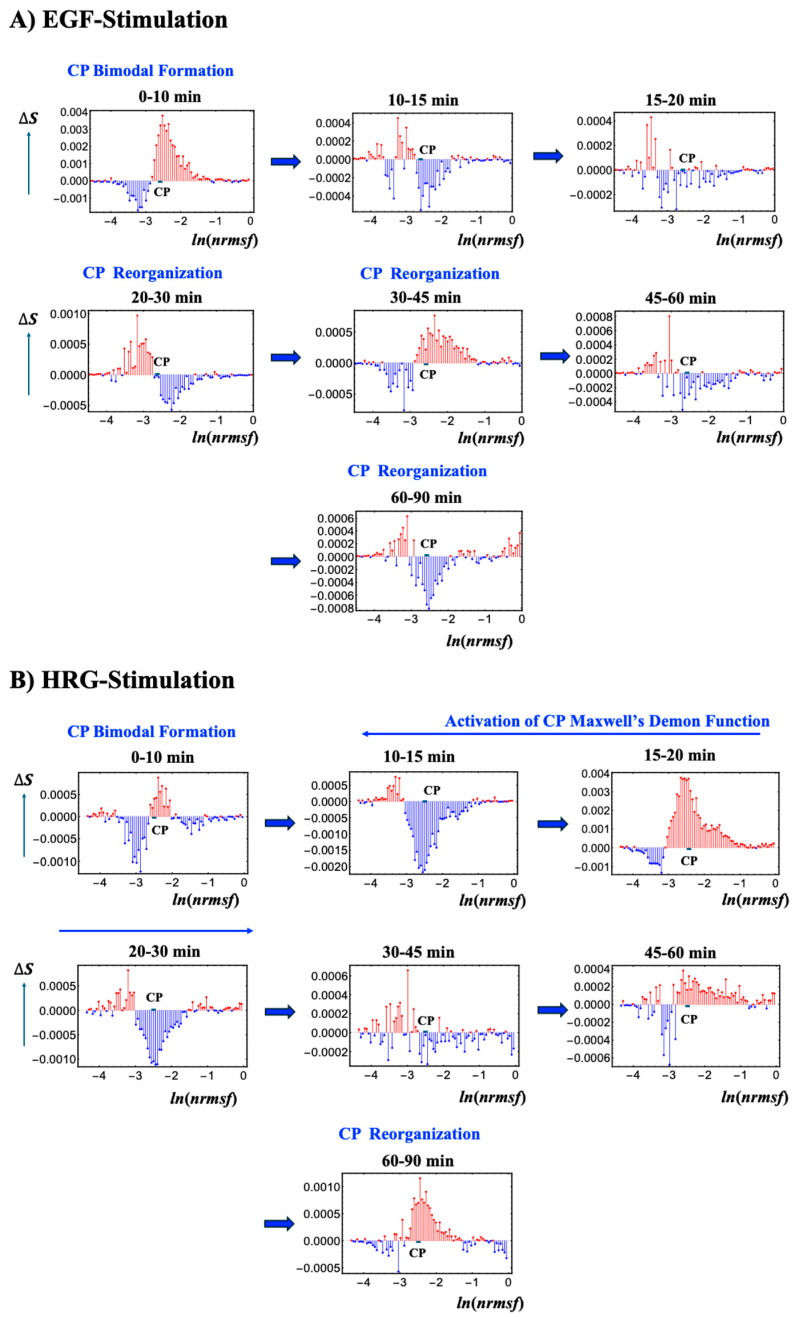
**Chromatin dynamics and CP bimodal formation across temporal entropy changes**. Temporal changes in Shannon entropy, reflecting chromatin dynamics, are plotted against the natural logarithm of *nrmsf* (80 bins), *ln(nrmsf)* in EGF- (**A**) and HRG-stimulated (**B**) MCF-7 cells. The CP region, marked by the bold black solid line, displays singular bimodal behavior within the ranges −2.64 < *ln(nrmsf)* < −2.52 for EGF stimulation and −2.55 < *ln(nrmsf)* < −2.44 for HRG stimulation (see Figure 2). Under EGF stimulation, intermittent CP bimodal patterns emerge at specific intervals (e.g., 0–10 and 20–30 min) but repeatedly dissolve without triggering a critical transition, indicating that coherent, global chromatin remodeling does not occur, resulting in no cell fate change. In contrast, HRG stimulation initially produces a CP bimodal pattern during the 0–10 min period, which then transitions into a global, coherent fold-unfold chromatin change between 10 and 30 min. This is followed by the re-formation of CP states between 60 and 90 min, aligning with a cell fate transition (see Section 2.6). These results highlight cyclical chromatin organization states driven by CP–PES phase synchronization and thermodynamic interactions, including Maxwell’s demon-like chromatin memory effects (see further detail in Figure 9).

**Figure 9 ijms-26-04911-f009:**
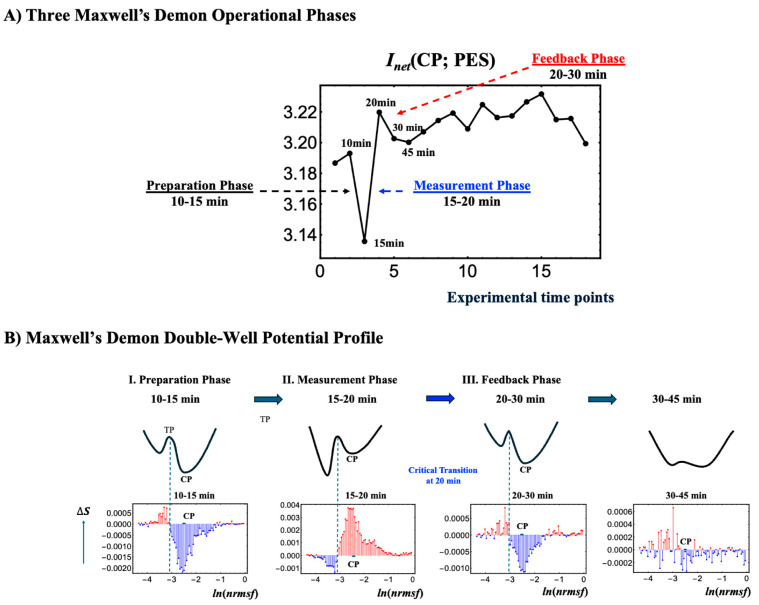
**Three Maxwell’s demon processes and schematic representation of its double-well potential profile**. (**A**) Net mutual information *I*_net_ (CP; PES) in HRG stimulation reveals all three operational phases of a Maxwell’s demon: preparation (10–15 min), measurement (15–20 min), and feedback (20–30 min). (**B**) This figure illustrates a conceptual double-well potential landscape representing chromatin dynamics, highlighting the Maxwell’s demon-like behavior of the CP within the range −2.55 < *ln(nrmsf)* < −2.44 (shown by thick solid black lines). Each well corresponds to a distinct chromatin state, such as folded or unfolded chromatin, separated by a potential barrier. The transition point (TP) occurs around *ln(nrmsf)* = −3.15 to −3.00, marking where chromatin restructuring takes place. The red line indicates increases in entropy and the blue line indicates decreases, reflecting chromatin remodeling dynamics—the oscillatory behavior of coherent chromatin unfolding (red) and folding (blue), dynamically centered around the CP. Coherent chromatin unfolding dynamics exhibit a pronounced peak at 15–20 min, which mirrors the first chromatin-unfolding step of pericentric-associated domains (PADs) under HRG stimulation [16]. After 30 min, the potential barrier almost disappears, suggesting an inactive Maxwell’s demon function with a loss of coherent chromatin state. (**A**) The *x*-axis represents experimental time points, and the *y*-axis represents the net mutual information value. (**B**) The *y*-axis represents changes in Shannon entropy associated with chromatin remodeling, and the *x*-axis represents the value of *ln(nrmsf)*.

### 2.7. Structural Signatures of Chromatin Remodeling in HRG-Stimulated MCF-7 Cells

Here, we provide further experimental evidence supporting the use of gene expression variability, quantified by *ln(nrmsf)* values, as a proxy for chromatin remodeling dynamics.

Chromatin folding and unfolding are highly complex biochemical processes, underscored by the enormous challenge of compressing approximately 2 m of human DNA into the few-micrometer space of a cell nucleus. The “structural signatures” of chromatin remodeling were revealed through the combined application of confocal microscopy image analysis and biochemical approaches [16]. In their study, Krigerts and colleagues focused on pericentric-associated domains (PADs) and chromocentres, higher-order complexes formed by the aggregation of PADs.

Chromocentres serve as markers of chromatin’s folded or unfolded state, co-segregating with repressive genomic regions and contributing to gene silencing near centromeres via position effect variegation [57,58,59]. PADs—and consequently, chromocentres—form through the transient aggregation of histones and protamines, and are characterized by a highly variable acetylation pattern [58]. The relative size of chromocentres determines their effect on chromatin density. A single PAD has an approximate area of 1 µm^2^. Chromocentres composed of fewer PADs indicate lower chromatin density, which is associated with increased gene expression variability.

In the early phase of the cell fate transition in HRG-stimulated cells, corresponding to 15–30 min after drug administration, a marked disaggregation of chromocentres is observed, giving rise to isolated PADs. This structural transition aligns with the critical transition described in Section 2.4, Section 2.5 and Section 2.6. Furthermore, as expected in an aggregation/disaggregation process, the number of chromocentres follows a power-law scaling with their relative size (see Figure 10).

**Figure 10 ijms-26-04911-f010:**
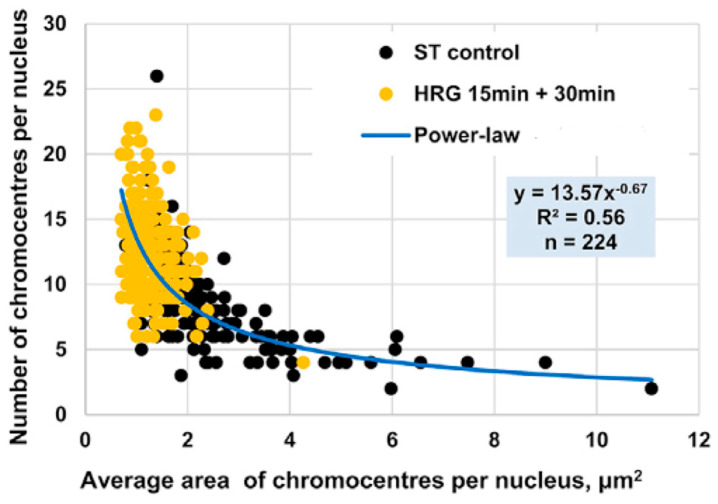
**Power-Law Scaling of Chromocentre Distribution in MCF-7 Cells Under HRG Stimulation**. Area (*x*-axis) and number of PADs (*y*-axis) in MCF-7 cells as observed using confocal microscopy. Black dots represent PADs in untreated (ST control) cells, while yellow dots indicate PADs under HRG stimulation during the transition phase. The distribution follows a power-law scaling: small chromocentres significantly outnumber larger ones, with most chromocentres comprising a single PAD during the critical transition.

During the transition phase, chromocentres disaggregate into single PADs, corresponding to targeted chromatin unfolding and a consequent increase in gene expression variability. Figure 9B consistently shows that chromatin unfolding dynamics reach the first peak at 15–20 min [16]. A coherent folding phase emerges at 20–30 min and fades by 30–45 min. The associated entropy changes are about an order of magnitude smaller than those of the unfolding peak. The dynamics of PADs provide proof-of-concept for the hypothesis that temporal gene expression variability by *ln(nrmsf)* value is a functional consequence of chromatin remodeling (see Section 2.6). This notion is further supported by the observed upregulation of key differentiation master genes such as FOS and c-Myc [16]. 

Together, these findings establish a crucial link between our statistical mechanics framework, which views the genome as a unified system whose dynamics are reflected in the global distribution of gene expression variability, and the classical, single-gene perspective of molecular biology (see Section 3).

As a conclusive remark, our information thermodynamics analysis (ITA) demonstrates that the CP functions as a Maxwell’s demon in genome regulation by driving cell-fate change, as supported by the following four independent lines of evidence:It fulfills all three characteristic operational phases of a Maxwell’s demon [11,30,31] (see Section 2.5);Experimental and computational analyses of chromatin dynamics based on *ln(nrmsf)*-sorted whole-genome expression data consistently reveal regulatory patterns indicative of Maxwell’s demon–like behavior (Section 2.6 and Section 2.7);These findings are confirmed by an independent replicate of the gene expression dataset of HRG-stimulated MCF-7 cells (see Section 4) as detailed in the Appendix A.Similar regulatory behavior observed in a different cancer cell type, dimethyl sulfoxide (DMSO)-stimulated HL-60 cells [23], as detailed in the Appendix A, further reinforces this conclusion.

## 3. Discussion: Biophysical Implications

Our findings highlight the genome as an open thermodynamic system operating far from equilibrium, where continuous energy dissipation sustains dynamic chromatin organization and gene expression. This dissipative framework explains how cells preserve functional stability amid environmental fluctuations.

From a biomedical standpoint, this thermodynamic perspective has substantial implications. By elucidating the principles governing how cells adopt and maintain distinct fates, we gain insights into processes such as cancer progression, metastasis, and therapeutic resistance. Targeting the CP hub may therapeutically reprogram cell states by altering chromatin remodeling, gene expression, signaling, epigenetics, cell cycle and metabolism. For instance, restricting undesirable plasticity in cancer cells or guiding stem cells toward more favorable differentiation pathways could offer new strategies for improved therapeutic interventions.

In Section 3, we further expand our findings and discuss the potential for controlling the dynamics on the fate of cancer cells in three key aspects: (1) order parameter of phase synchronization, (2) autonomous genome computing, and (3) genome intelligence (GI).

**(1)** 
**Existence of an Order Parameter for Phase Synchronization by Integrating Dynamical System Analysis and Non-Equilibrium Thermodynamics**


We explore the emergence of an order parameter for phase synchronization by integrating the genome engine mechanism, a dynamical systems approach based on expression flux analysis [7,8,9], with genomic thermodynamic phase synchronization, a nonequilibrium thermodynamic approach. This integration is important because it suggests a biophysical link between the principles governing dynamical systems and the mechanisms of thermodynamics in genome regulation.

Under HRG stimulation, during the critical transition, the CP undergoes a substantial change in all information-thermodynamic metrics, including entropy, both at the CP and in the PES (whole expression system without the CP genes) (Figure 5B). Additionally, there is a marked increase in self-flux (effective force) at both the CP and the genome attractor (GA) (Figure 11A). This critical transition induces a drastic system-wide shift, where all statistical measures/parameters describing the state become highly correlated, effectively collapsing the dynamical space into a binary regime: CP activation (HRG) vs. no CP activation (EGF). Consequently, every computed index converges under a unified critical framework, suggesting that the dynamics of phase transitions not only drive biological coherence (pulse phase in biphasic induction of AP-1 complex [55]) but also integrate diverse analytical metrics into a single, self-operating system.

Figure 11 provides direct evidence of this collapse, showing that the CP entropy change (Δ*S*(CP)), representing thermodynamic behavior, synchronizes with the temporal evolution of self-flux (effective force) at both the CP and the genome attractor (GA), thereby guiding large-scale changes in genomic expression dynamics (see Section 1). This concurrent synchronization bridges thermodynamic and dynamical system behaviors. Formerly independent statistical descriptors become tightly correlated, and simultaneous surges in Δ*S*(CP), CP self-flux, and GA self-flux confirm that thermodynamic and dynamical cues act synergistically to enforce genome-wide phase coherence.

The convergence of different descriptors suggests the emergence of a robust global order parameter, analogous to the Kuramoto measure [60,61]. This scalar measure ranges from near-zero (indicating incoherent or random phase relationships) to one (denoting near-perfect synchrony). The order parameter thus encapsulates the degree of phase alignment among the CP and GA, serving as a macroscopic indicator of the (critical) phase transition. As the CP and GA become increasingly synchronized, the order parameter rises sharply, marking the transition from a disordered to an ordered state—a signature hallmark of self-organized critical (SOC) control in whole expression. Thus, this pulse-like phase transition not only underscores the CP’s role as the central regulatory hub in cancer cell expression but also bridges dynamical and statistical perspectives, unifying diverse analytical metrics into a single, self-operating system.

**Figure 11 ijms-26-04911-f011:**
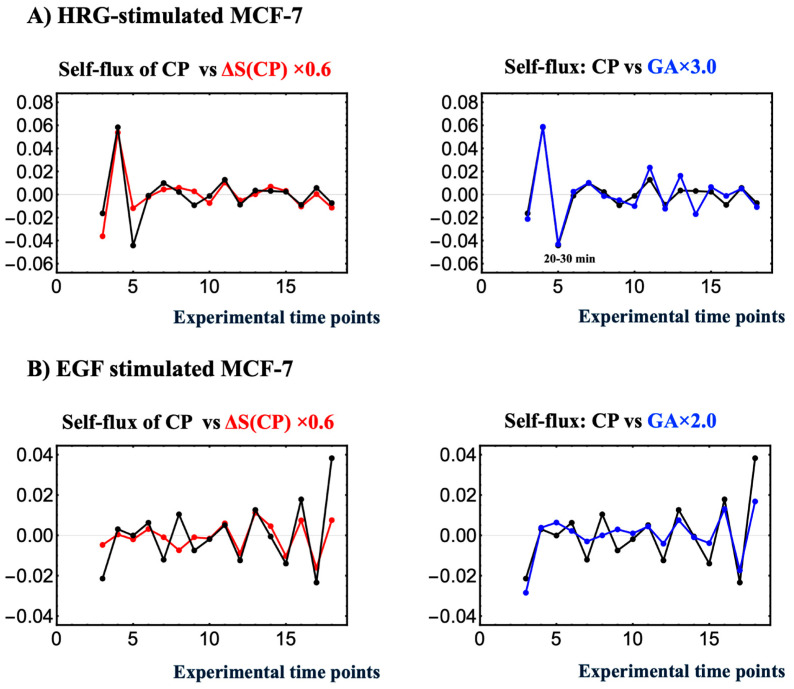
**Concurrent occurrence of entropy-based thermodynamic synchronization and force-like dynamical synchronization**. In HRG stimulation (**A**) and EGF stimulation (**B**), the left panel shows the phase synchrony between self-flux (a dynamical system measure) and entropy change Δ*S*(CP) (a thermodynamic measure) for CP genes, while the right panel presents the self-flux of the CP and genome attractor (GA). Here, the self-flux of the CP and GA is defined as the second finite difference from their overall temporal average value: −(*ε*(*t_j_*_+1_) − 2*ε*(*t_j_*) + *ε*(*t_j_*_−1_)), where *ε*(*t_j_*) represents the expression of the center of the mass (assuming unit mass) of the CP genes or GA at *t* = *t_j_*. The negative sign arises from the correspondence with a harmonic oscillator when the system becomes linear. The change in CP entropy is given by its first finite difference: *S*(CP(*t_j_*)) − *S*(CP(*t_j_*_−1_)). Phase synchrony becomes apparent after the 10-min interval, with Δ*S*(CP) after 10–15 min and self-flux after 0–10–15 min. HRG stimulation, which induces a critical transition, shows stronger phase synchrony compared to EGF stimulation, where no critical transition occurs. In contrast, in the EGF case, the early-time CP formation–deformation cycle suggests that it leads to weak phase synchrony among CP self-flux, its entropy change, and GA self-flux (see Figure 8A).

**(2)** 
**Autonomous Genome Computing as a Conceptual Framework**


Our study demonstrates that net mutual information, incorporating higher-order nonlinear interactions, drives the critical transition that guides cell fate. This transition is governed by genomic-thermodynamic phase synchronization and is mediated by the CP genes functioning as a rewritable chromatin memory. This framework grounds the concept of autonomous genome computing in measurable, mechanistic processes, where the genome integrates regulatory inputs and dynamically transitions between distinct functional states.

Recent advances suggest that genomic DNA is not merely a static blueprint but rather a dynamic, computationally capable system—a concept we term “genome computing”. In this framework, “computation” describes the genome’s autonomous ability to integrate diverse regulatory inputs, store adaptive information, and transition between functional states in a context-dependent manner. This perspective unifies our understanding of the nonlinear and adaptive behaviors that regulate gene expression, drive cell differentiation, and govern cellular responses to external cues.

One key insight is that time-dependent transitions in genomic states can be modeled using kinetic equations with cubic nonlinearity, derived from symmetry arguments in thermodynamics [62,63]. This nonlinearity arises from changes in translational and conformational entropy within large DNA molecules interacting with their associated ionic environments [64,65,66,67,68]. Experimentally observed chromatin reorganizations support this perspective, suggesting that genomic architecture is continuously tuned to explore and settle into favorable configurations.

The symmetry characterized by cubic nonlinearity mirrors excitability phenomena observed in neuronal systems, as described by the FitzHugh–Nagumo [69,70] and Hodgkin–Huxley [71] frameworks. While these parallels remain conceptual, they underscore a shared principle: nonlinear systems, whether neural or genomic, can exhibit threshold-dependent switching between stable states.

As demonstrated in Section 2.6, the CP region exhibits large-scale bimodal singular behavior based on *ln(nrmsf)* values. Consistent with this, recent evidence on bimodality clearly demonstrates that genome-sized DNA exhibits a bimodal free energy profile [64,67,72,73,74,75,76]. This bimodal symmetry inherently implies cubic nonlinearity in the reaction kinetics, which arises from the functional derivative of the free energy [62,63,77,78].

At the heart of these transitions lies the occurrence of the critical point (CP). While its precise composition remains to be fully delineated, existing evidence suggests that specific genomic regions consistently orchestrate large-scale regulatory shifts. As cells approach critical decision points such as lineage commitment, the CP emerges from bimodal states, toggling chromatin configurations between more and less active forms. Observations of pericentromere-associated domain (PAD) rearrangements and chromatin remodeling [16] further support the notion that the CP governs these critical transitions, effectively “choosing” among distinct genomic states.

These transitions can be conceptualized through an energy landscape perspective, where changes in translational and conformational entropy reshape the free energy landscape, favoring certain pathways over others. Analogies with Ginzburg–Landau theory [79] highlight the importance of identifying order parameters (e.g., chromatin compaction or gene network states) and recognizing that critical transitions occur when specific thresholds are crossed. Our self-organized criticality (SOC) provides another useful framework: the genome, via the CP, naturally tunes itself to critical points, maintaining a balance between order and flexibility. Acting as rewritable chromatin memory, the CP plays a role akin to Maxwell’s demon, selectively promoting transitions that reduce uncertainty and drive the system toward coherent and functionally relevant states.

This notion of “genome computing” does not imply that the genome functions like a digital processor with discrete inputs and outputs. Instead, genome computing is characterized by autonomous, decentralized super-information processing. It reflects a continuous, adaptive reshaping of genomic states, where “memory” is encoded in the genome’s ability to revisit specific conformations, and “logic” emerges from network interactions that drive transitions toward or away from stable regulatory configurations.

In conjunction with the genomic-thermodynamic mechanism, understanding how the CP functions as a central process hub to orchestrate the complex spatiotemporal self-organization of the genome will elucidate fundamental principles governing cell fate, development, and stress responses. This knowledge should, ultimately, inspire innovative approaches in regenerative medicine, cancer therapy, and synthetic biology.

**(3)** 
**Toward Genome Intelligence: A Future Perspective**


At a more theoretical level, the concept of genome intelligence (GI) builds upon the computational principles of genome computing to describe the genome’s capacity for integrated, adaptive, and memory-like behavior. Here, intelligence refers to a minimal form of integrated responsiveness, where the genome not only reacts to environmental stimuli but incorporates these inputs into its regulatory framework, enabling adaptation and future decision-making.

Integrated information theory (IIT), originally developed to explain consciousness in neural systems [80], offers a useful framework for understanding intelligence in biological systems more broadly. The fundamental challenge addressed by IIT is how to integrate external informative stimuli into the structure of an “intelligent agent”, leading to the emergence of a memory of the stimulus. This stands in contrast to non-intelligent sensors (e.g., a photodiode), whose structure remains unchanged by interactions with stimuli.

As demonstrated by Niizato et al. (2024) [81], IIT extends beyond neural systems to encompass molecular and cellular systems, where complex interactions give rise to irreducible wholes—systems that cannot be decomposed into independent parts without loss of function. When applied to the genome, GI emerges from the integration of external signals into chromatin configurations, regulatory networks, and gene expression states. This capacity for integration, persistence, and reconfiguration distinguishes the genome as an autonomous, intelligent system.

It is important to clarify that referring to GI does not imply a claim that the genome possesses intelligence in the fully defined, cognitive sense. The term “intelligence” derives from the Latin *intus* + *legere*, meaning “to read within”-that is, to uncover hidden or implicit knowledge. While the genome lacks intentional insight, referring to genome intelligence emphasizes its ability to track past experiences, as demonstrated by epigenetic memory [82].

Ultimately, GI reframes our understanding of the genome as a dynamic, adaptive, and integrative system. In our view, GI highlights the consilience of computational and thermodynamic principles with biological mechanisms, offering a foundation for exploring how genomes encode past experiences, adapt to environmental changes, and guide cellular behavior. This perspective has profound implications for fields such as regenerative medicine and synthetic biology, where leveraging the intelligence-like properties of the genome may transform how we design and manipulate living systems.

## 4. Materials

Microarray data for the activation of ErbB receptor ligands in human breast cancer MCF-7 cells by EGF and HRG; Gene Expression Omnibus (GEO) ID: GSE13009 (*N* = 22,277 mRNAs; for experimental details see [22]) at 18 time points: t_1_ = 0, t_2_ = 10, 15, 20, 30, 45, 60, 90 min, 2, 3, 4, 6, 8, 12, 24, 36, 48, t_T_ = 18 = 72 h. Each condition includes two replicates (rep 1 and rep 2); the analyses presented in this report are based on rep 1 for both EGF and HRG, while the results from rep 2 of HRG stimulation are provided in the Appendix A. The robust multichip average (RMA) was used to normalize expression data for further background adjustment and to reduce false positives [83,84,85].

## 5. Conclusions

Our study demonstrates that genomic regulation can be understood through the lens of open stochastic thermodynamics. Rather than existing at thermodynamic equilibrium, gene expression operates far from equilibrium, with continuous energy dissipation enabling dynamic regulation, adaptability, and responsiveness to external cues. By examining the roles of critical point (CP) genes and the whole expression system (WES) in MCF-7 breast cancer cells, we present a thermodynamic framework that explains how biological systems autonomously maintain coherence, stability, and flexibility in their gene expression profiles.

Key insights from our findings include the thermodynamic phase synchronization of CP genes with the genome, the distinct roles of the CP under different stimuli, and the presence of positive higher-order mutual information. The latter highlights nonlinear interdependencies, enhancing both synergy (emergent collective information) and redundancy (overlapping information) beyond the framework of the pairwise framework of interactions among genes.

Under epidermal growth factor (EGF) stimulation, the CP remains passive, preserving stability without transitioning into a new state. In contrast, under heregulin (HRG) stimulation, the CP “reads” and redistributes information through the rewritable chromatin memory, exhibiting a Maxwell’s demon-like function to orchestrate a global shift in gene expression. In other words, the genome “learns” from environmental signals, processes that information, and then reprograms its overall state—a phenomenon that can be described as “Genome Intelligence (GI)”.

On a broader level, these computational dynamics support the concept of GI, where the genome exhibits emergent properties such as the ability to discriminate between environmental states, integrate signals into its structural framework, and adaptively guide future regulatory transitions. This emergent intelligence reflects the genome’s capacity to store “memory” of past states and utilize this information to navigate complex decision-making landscapes. These insights bridge classical molecular genetics with nonequilibrium thermodynamics and computational principles, providing a holistic view of genome regulation.

Our findings open new avenues for understanding disease progression, guiding regenerative medicine, and informing innovative cancer therapies. For example, the CP’s ability to dynamically “compute” and integrate signals highlights opportunities for targeting critical regulatory hubs in cancer treatment or leveraging genomic intelligence in synthetic biology to design adaptive, programmable cellular systems.

From a global perspective, this integration of genome computing and intelligence inspires the development of machine learning tools modeled on cell fate transition dynamics. Physics-Informed Neural Networks (PINNs), as described by Cai et al. (2021) [86], offer a promising framework for incorporating physical principles into neural network architectures to replicate the behavior of biological systems. Similarly, Hopfield networks, as noted by Krotov [87], demonstrate how network correlations can encode emergent behavior, much like genomic systems encode adaptive regulatory transitions. By applying these computational principles as hyper-parallel autonomous decentralized systems, we can enhance both biological understanding and machine learning, paving the way for models that seamlessly integrate thermodynamic, computational, and biological insights.

## Data Availability

The data used in this study are publicly available from the Gene Expression Omnibus (GEO) under accession ID GSE13009.

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
