# Peer review of "Genomic-Thermodynamic Phase Synchronization: Maxwell’s Demon-like Regulation of Cell Fate Transition"

_ijms, 2025, doi:10.3390/ijms26104911_

Round 1

Reviewer 1 Report (New Reviewer)

Comments and Suggestions for Authors

This study presents a novel biophysical framework integrating thermodynamics, information theory, and genomic regulation to elucidate cell fate transitions in MCF-7 breast cancer cells. The authors propose that the Critical Point (CP), a subset of genes with bimodal expression variability, acts as a Maxwell’s demon-like regulatory hub, orchestrating genome-wide phase synchronization and critical transitions. The work is conceptually innovative, bridging nonequilibrium thermodynamics with biological regulation, and offers potential implications for cancer research and cell reprogramming. However, methodological clarity, data interpretation, and structural organization require refinement to strengthen the manuscript’s impact.

Major

1.The calculation of nrmsf and criteria for gene grouping (e.g., thresholds for "critical states") are insufficiently detailed. Clarify how bin sizes, bootstrapping parameters, and convergence criteria were chosen.

2.The distinction between "logical" and "physical" states in entropy decomposition (Figure 7) needs a more intuitive explanation.

3.Figures 5–8 lack sufficient annotations. For example, the temporal correlation between CP entropy and net mutual information (Figure 6) should be explicitly tied to biological mechanisms (e.g., chromatin remodeling).

4.The claim that "CP acts as Maxwell’s demon" requires stronger experimental validation. While entropy changes align with this analogy, direct evidence of energy-information tradeoffs (e.g., thermodynamic costs of CP activity) is missing.

5.The sections on "Genome Intelligence" and "Autonomous Genome Computing" extend beyond the scope of the data. While intriguing, these ideas are not sufficiently grounded in the results and risk diluting the manuscript’s focus.

Minor

1.Provide a step-by-step workflow for nrmsf calculation, gene clustering, and entropy estimation. Include pseudocode or equations for bootstrapping procedures.

2.Clarify how the "logical state" framework (Figure 7) differs from traditional entropy decomposition.

3.Annotate key time points in Figures 5–8 (e.g., AP-1 activation, c-MYC peaks) to link thermodynamic metrics to known biological events.

4.Include error bars or confidence intervals in entropy/mutual information plots (e.g., Figure 4) to assess statistical robustness.

5.Reorganize the Discussion to prioritize mechanistic insights over theoretical extrapolations.

Comments on the Quality of English Language

Simplify overly complex sentences (e.g., "genomic avalanche") and avoid repetitive descriptions of phase synchronization.

Author Response

Reviewer 2 Report (New Reviewer)

Comments and Suggestions for Authors

This manuscript presents an intriguing and original perspective that is rarely explored in the biological sciences

The study adopts a conceptual framework common to physics and information theory, which differs significantly from the approaches commonly employed in biological and health sciences. As a result, the presentation and interpretation of the data may be challenging for the journal’s primary audience of biologists and health professionals to follow or evaluate. In that sense, the assessment of the scientific content of such a manuscript will always be biased by the reviewer's background. As a biologist/health scientist, I will focus my reviews on the biological aspects and refrain from commenting on the work's theoretical physical elements.

An essential aspect of biology is the immense number of intrinsic and extrinsic variables that bring enormous variability to systems. Unlike in the physical sciences, initial states in biology are never truly identical, and replication never perfectly recapitulates prior conditions. So, applying frameworks that assume precise initial conditions may oversimplify the nature of biological systems and overlook context-specific variability. This is such a critical point in this manuscript. The authors interpret the temporal variability in gene expression as evidence for the non-deterministic nature of genome regulation. However, it is difficult to distinguish true stochasticity from external noise or technical variability in biology.

For instance, It is not clear whether technical and biological sources of variation (e.g., cell cycle heterogeneity, metabolic differences, or culture conditions) were adequately accounted for and whether they can have a role in the presented results. Clear documentation of experimental conditions and analysis pipelines would help diminish this questioning type. For instance, were the cells in the exact cell cycle phase? Do they have the same confluence, metabolic activity, and passage number? How about incubation and treatment times? Were they precisely the same? However, even with the most rigorous experimental design, there are inherent limitations in how precisely we can control biological systems, especially living cells. In this way, when the authors attribute entropy and information flow patterns to “non-determinism,” they may be underestimating the role of unmeasured extrinsic variability — stemming from both the biological system and the experimenter. I don’t believe that is a flaw in the presented study. Even if a study aims for identical conditions, there’s always residual variation, much of which is unmeasurable or unrecognized. This is a fundamental feature of biology. However, it is not clear how this variability impacts the presented results.

It is also important to highlight that nucleic acid sequencing technologies are fundamentally probabilistic. Each sequencing step introduces uncertainty that can affect the final expression values, from transcript capture and amplification to sequencing and quantification. In this way, sequencing data should not be treated as absolute representations of biological states but rather as statistically inferred approximations that are subject to noise and technical biases. Interpreting these data should take into account their probabilistic origin and associated uncertainty.

While the manuscript introduces a sophisticated thermodynamics-inspired perspective on gene expression regulation, it tends to overlook or minimize the inherent biological complexity of living systems. By abstracting gene expression dynamics into mathematically tractable models, the authors risk underestimating critical layers of biological regulation that cannot be easily captured through entropy metrics or phase synchronization alone. This simplification may limit the model's capacity to reflect a real-world biological system.

Additionally, this manuscript introduces theoretical concepts and frameworks unfamiliar to this reviewer, which raises the possibility that some (or all) of my comments may reflect misunderstandings or misinterpretations. However, this points to a broader issue: the manuscript presents its central ideas in a way that may be inaccessible to much of the journal’s intended audience of biologists and health scientists. By relying heavily on terminology and logic derived from physics without sufficient biological grounding or explanatory bridge, the authors risk limiting the reach and impact of their findings. While interdisciplinary approaches are vital for innovation, effectively communicating across disciplinary boundaries requires attention to clarity, context, and the field's conventions. In its current form, the manuscript may unintentionally come across as disconnected from the scientific culture and expectations of the biological research community.

Author Response

This manuscript is a resubmission of an earlier submission. The following is a list of the peer review reports and author responses from that submission.

Round 1

Reviewer 1 Report

Comments and Suggestions for Authors

The article by Masa Tsuchiya and colleagues shows a new interpretation of gene expression regulation based on Maxwell’s Demon-Like Regulation of Cell Fate Transition using MCF-7 breast cancer cell line.  This study use data from Microarray data for the activation of ErbB receptor ligands in human breast cancer MCF-7 cells by EGF and HRG; Gene Expression Omnibus (GEO) ID: GSE13009 (N = 22277 938) and many mathematics methods to illustrate the temporal changes in Shannon entropy associated with chromatin dynamics and gene expression. The authors conclude that critical distinction between HRG and EGF stimulation lies in the activation of the CP (critical point) as a Maxwell demon under HRG by analyzing entropy variations within the system. The literature presented in the article is updated and it seems that the conclusion of the authors is correct, although there are many points that deserve further investigation such as types of cell synchronization used in the experience, evaluation of phase of cell cycle and profile of epigenetic memory, etc.